# AMBERT: A PRE-TRAINED LANGUAGE MODEL WITH MULTI-GRAINED TOKENIZATION

## ABSTRACT

Pre-trained language models such as BERT have exhibited remarkable performances in many tasks in natural language understanding (NLU). The tokens in the models are usually fine-grained in the sense that for languages like English they are words or sub-words and for languages like Chinese they are characters. In English, for example, there are multi-word expressions which form natural lexical units and thus the use of coarse-grained tokenization also appears to be reasonable. In fact, both fine-grained and coarse-grained tokenizations have advantages and disadvantages for learning of pre-trained language models. In this paper, we propose a novel pre-trained language model, referred to as AMBERT (A Multi-grained BERT), on the basis of both fine-grained and coarse-grained tokenizations. For English, AMBERT takes both the sequence of words (fine-grained tokens) and the sequence of phrases (coarse-grained tokens) as input after tokenization, employs one encoder for processing the sequence of words and the other encoder for processing the sequence of the phrases, utilizes shared parameters between the two encoders, and finally creates a sequence of contextualized representations of the words and a sequence of contextualized representations of the phrases. Experiments have been conducted on benchmark datasets for Chinese and English, including CLUE, GLUE, SQuAD and RACE. The results show that AMBERT outperforms the existing best performing models in almost all cases, particularly the improvements are significant for Chinese. We also develop a version of AMBERT which performs equally well as AMBERT but uses about half of its inference time.

## 1 INTRODUCTION

Pre-trained models such as BERT, RoBERTa, and ALBERT (Devlin et al., 2018; Liu et al., 2019; Lan et al., 2019) have shown great power in natural language understanding (NLU). The Transformer-based language models are first learned from a large corpus in pre-training, and then learned from labeled data of a downstream task in fine-tuning. With Transformer (Vaswani et al., 2017), pre-training technique, and big data, the models can effectively capture the lexical, syntactic, and semantic relations between the tokens in the input text and achieve the state-of-the-art performances in many NLU tasks, such as sentiment analysis, text entailment, and machine reading comprehension.

In BERT, for example, pre-training is mainly conducted based on mask language modeling (MLM) in which about 15% of the tokens in the input text are masked with a special token [MASK], and the goal is to reconstruct the original text from the masked text. Fine-tuning is separately performed for individual tasks as text classification, text matching, text span detection, etc. Usually, the tokens in the input text are fine-grained; for example, they are words or sub-words in English and characters in Chinese. In principle, the tokens can also be coarse-grained, that is, for example, phrases in English and words in Chinese. There are many multi-word expressions in English such as 'New York' and 'ice cream' and the use of phrases also appears to be reasonable. It is more sensible to use words (including single character words) in Chinese, because they are basic lexical units. In fact, all existing pre-trained language models employ single-grained (usually fine-grained) tokenization.

Previous work indicates that the fine-grained approach and the coarse-grained approach have both pros and cons. The tokens in the fine-grained approach are less complete as lexical units but their representations are easier to learn (because there are less token types and more tokens in training data), while the tokens in the coarse-grained approach are more complete as lexical units but their

representations are more difficult to learn (because there are more token types and less tokens in training data). Moreover, for the coarse-grained approach there is no guarantee that tokenization (segmentation) is completely correct. Sometimes ambiguity exists and it would be better to retain all possibilities of tokenization. In contrast, for the fine-grained approach tokenization is carried out at the primitive level and there is no risk of 'incorrect' tokenization.

For example, Li et al. (2019) observe that fine-grained models consistently outperform coarse-grained models in deep learning for Chinese language processing. They point out that the reason is that low frequency words (coarse-grained tokens) tend to have insufficient training data and tend to be out of vocabulary, and as a result the learned representations are not sufficiently reliable. On the other hand, previous work also demonstrates that masking of coarse-grained tokens in pre-training of language models is helpful (Cui et al., 2019; Joshi et al., 2020). That is, although the model itself is fine-grained, masking on consecutive tokens (phrases in English and words in Chinese) can lead to learning of a more accurate model. In Appendix A, we give examples of attention maps in BERT to further support the assertion.

In this paper, we propose A Multi-grained BERT model (AMBERT), which employs both fine-grained and coarse-grained tokenizations. For English, AMBERT extends BERT by simultaneously constructing representations for both words and phrases in the input text using two encoders. Specifically, AMBERT first conducts tokenization at both word and phrase levels. It then takes the embeddings of words and phrases as input to the two encoders. It utilizes the same parameters across the two encoders. Finally it obtains a contextualized representation for the word and a contextualized representation for the phrase at each position. Note that the number of parameters in AMBERT is comparable to that of BERT, because the parameters in the two encoders are shared. There are only additional parameters from multi-grained embeddings. AMBERT can represent the input text at both word-level and phrase-level, to leverage the advantages of the two approaches of tokenization, and create richer representations for the input text at multiple granularity.

We conduct extensive experiments to make comparison between AMBERT and the baselines as well as alternatives to AMBERT, using the benchmark datasets in English and Chinese. The results show that AMBERT significantly outperforms single-grained BERT models with a large margin in both Chinese and English. In English, compared to Google BERT, AMBERT achieves 2.0% higher GLUE score, 2.5% higher RACE score, and 5.1% more SQuAD score. In Chinese, AMBERT improves average score by over 2.7% in CLUE. Furthermore, a simplified version AMBERT with only the fine-grained encoder can preform much better than the single-grained BERT models with a similar amount of inference computation.

We make the following contributions in this work.

- Study of multi-grained pre-trained language models,
- Proposal of a new pre-trained language model called AMBERT as an extension of BERT, which makes use of multi-grained tokens and shared parameters,
- Empirical verification of AMBERT on the English and Chinese benchmark datasets GLUE, SQuAD, RACE, and CLUE.

## 2 RELATED WORK

There has been a large amount of work on pre-trained language models. ELMo (Peters et al., 2018) is one of the first pre-trained language models for learning of contextualized representations of words in the input text. Leveraging the power of Transformer (Vaswani et al., 2017), GPTs (Radford et al., 2018; 2019) are developed as unidirectional models to make prediction on the input text in an auto-regressive manner, and BERT (Devlin et al., 2018) is developed as a bidirectional model to make prediction on the whole or part of the input text. Mask language modeling (MLM) and next sentence prediction (NSP) are the two tasks in pre-training of BERT. Since the inception of BERT, a number of new models have been proposed to further enhance the performance of it. XLNet (Yang et al., 2019) is a permutation language model which can improve the accuracy of MLM. RoBERTa (Liu et al., 2019) represents a new way of training more reliable BERT with a very large amount of data. ALBERT (Lan et al., 2019) is a light-weight version of BERT, which shares parameters across layers. StructBERT (Wang et al., 2019) incorporates word and sentence structures into BERT for learning of better representations of tokens and sentences. ERNIE2.0 (Sun et al., 2020) is a variant of BERT pre-trained in multiple tasks with coarse-grained tokens masked. ELECTRA (Clark et al., 2020) has a GAN-style architecture for efficiently utilizing all tokens in pre-training.

It has been found that the use of coarse-grained tokens is beneficial for pre-trained language models. Devlin et al. (2018) point out that 'whole word masking' is effective for training of BERT. It is also observed that whole word masking is useful for building a Chinese BERT (Cui et al., 2019). In ERNIE (Sun et al., 2019b), entity level masking is employed as a strategy for pre-training and proved to be effective for language understanding tasks (see also (Zhang et al., 2019)). In SpanBERT (Joshi et al., 2020), text spans are masked in pre-training and the learned model can substantially enhance the accuracies of span selection tasks. It is indicated that word segmentation is especially important for Chinese and a BERT-based Chinese text encoder is proposed with n-gram representations (Diao et al., 2019). All existing work focuses on the use of single-grained tokens in learning and utilization of pre-trained language models. In this work, we propose a general technique of exploiting multi-grained tokens for pre-trained language models and apply it to BERT.

## 3 OUR METHOD: AMBERT

In this section, we present the model, pre-training, and fine-tuning of AMBERT. We also make a discussion on alternatives of AMBERT.

### 3.1 MODEL

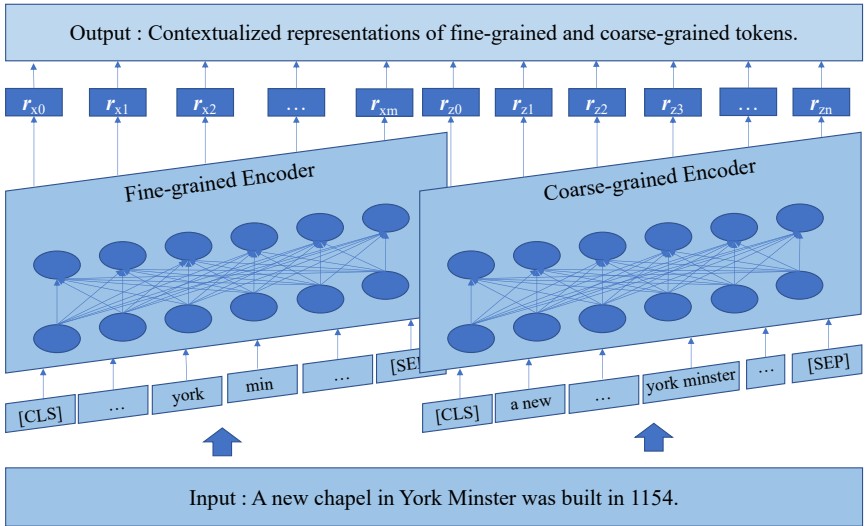

Figure 1: An overview of AMBERT, showing the process of creating multi-grained representations. The input is a sentence in English and output is the overall representation of the sentence. There are two encoders for processing the sequence of fine-grained tokens and the sequence of coarse-grained tokens respectively. The final contextualized representations of fine-grained tokens and coarse-grained tokens are denoted as $r_{x0}, r_{x1}, \cdots, r_{xm}$ and $r_{z0}, r_{z1}, \cdots, r_{zn}$ respectively.

Figure 1 gives an overview of AMBERT. AMBERT takes a text as input. Tokenization is conducted on the input text to obtain a sequence of fine-grained tokens and a sequence of coarse-grained tokens. AMBERT has two encoders, one for processing the fine-grained token sequence and the other for processing the coarse-grained token sequence. Each of the encoders has exactly the same architecture as that of BERT (Devlin et al., 2018) or Transformer encoder (Vaswani et al., 2017). The two encoders share the same parameters at each corresponding layer, except that each has its own token embedding parameters. The fine-grained encoder generates contextualized representations from the sequence of fine-grained tokens through its layers. In parallel, the coarse-grained encoder generates contextualized representations from the sequence of coarse-grained tokens through its layers. AMBERT outputs a sequence of contextualized representations for the fine-grained tokens and a sequence of contextualized representations for the coarse-grained tokens.

AMBERT is expressive in that it learns and utilizes contextualized representations of the input text at both fine-grained and coarse-grained levels. The model retains all possibilities of tokenizations and automatically learns the attention weights (importance) of representations of multi-grained tokens. AMBERT is also efficient through sharing of parameters between the two encoders. The parameters

represent the same ways of combining representations, no matter whether representations are those of fine-grained tokens or coarse-grained tokens.

## 3.2 PRE-TRAINING

Pre-training of AMBERT is mainly conducted on the basis of mask language modeling (MLM), at both fine-grained and coarse-grained levels. Next sentence prediction (NSP) is not essential as indicated in many studies after BERT (Lan et al., 2019; Liu et al., 2019). We only use NSP in our experiments for comparison purposes). Let $\hat{\mathbf{x}}$ denote the sequence of fine-grained tokens with some of them being masked, and $\bar{\mathbf{x}}$ denote the masked fine-grained tokens. Let $\hat{\mathbf{z}}$ denote the sequence of coarse-grained tokens with some of them being masked, and $\bar{\mathbf{z}}$ denote the masked coarse-grained tokens. Pre-training is defined as optimization of the following function,

$$\min_{\theta} \ -\log p_\theta(\bar{\mathbf{x}}, \bar{\mathbf{z}} | \hat{\mathbf{x}}, \hat{\mathbf{z}}) \approx \min_{\theta} -\sum_{i=1}^{m} m_i \log p_\theta(x_i | \hat{\mathbf{x}}) - \sum_{j=1}^{n} n_j \log p_\theta(z_j | \hat{\mathbf{z}}), \tag{1}$$

where $m_i$ takes 1 or 0 as values and $m_i = 1$ indicates that fine-grained token $x_i$ is masked, $m$ denotes the total number of fine-grained tokens; $n_j$ takes 1 or 0 as values and $n_j = 1$ indicates that coarse-grained token $z_j$ is masked, $n$ denotes the total number of coarse-grained tokens; and $\theta$ denotes parameters.

## 3.3 FINE-TUNING

In fine-tuning of AMBERT for classification, the fine-grained encoder and coarse-grained encoder create special [CLS] representations, and both representations are used for classification. Fine-tuning is defined as optimization of the following function, which is a regularized loss of multi-task learning, starting from the pre-trained model,

$$\min_{\theta} \ -\log p_\theta(\boldsymbol{y}|\mathbf{x}) = \min_{\theta} -\log p_\theta(\boldsymbol{y}|\boldsymbol{r}_{x0}) - \log p_\theta(\boldsymbol{y}|\boldsymbol{r}_{z0}) - \log p_\theta(\boldsymbol{y}|[\boldsymbol{r}_{x0}, \boldsymbol{r}_{z0}]) + \lambda \|\tilde{\boldsymbol{y}}_x - \tilde{\boldsymbol{y}}_z\|_2,$$
$$\tag{2}$$

where $\mathbf{x}$ is the input text, $\boldsymbol{y}$ is the classification label, $\boldsymbol{r}_{x0}$ and $\boldsymbol{r}_{z0}$ are the [CLS] representations of fine-grained encoder and coarse-grained encoder, $[\boldsymbol{a}, \boldsymbol{b}]$ denotes concatenation of vectors $\boldsymbol{a}$ and $\boldsymbol{b}$, $\lambda$ is coefficient, and $\|\|_2$ denotes L2 norm. The last term is based on agreement regularization (Brantley et al., 2019), which forces agreement between the predictions ($\tilde{\boldsymbol{y}}_x$ and $\tilde{\boldsymbol{y}}_z$).

Similarly, fine-tuning of AMBERT for span detection can be carried out, in which the representations of fine-grained tokens are concatenated with the representations of corresponding coarse-grained tokens. The concatenated representations are then utilized in the task.

## 3.4 ALTERNATIVES

We can consider two alternatives to AMBERT, which also rely on multi-grained tokenization. We refer to them as AMBERT-Combo and AMBERT-Hybrid and make comparisons of them with AMBERT in our experiments.

AMBERT-Combo has two individual encoders, an encoder (BERT) working on the fine-grained token sequence and the other encoder (BERT) working on the coarse-grained token sequence, without parameter sharing between them. In learning and inference AMBERT-Combo simply combines the output layers of the two encoders. Its fine-tuning is similar to that of AMBERT.

AMBERT-Hybrid has only one encoder (BERT) working on both the fine-grained token sequence and the coarse-grained token sequence. It creates representations on the concatenation of two sequences and lets the representations of the two sequences interact with each other at each layer. Its pre-training is formalized in the following function,

$$\min_{\theta} \ -\log p_\theta(\bar{\mathbf{x}}, \bar{\mathbf{z}} | \hat{\mathbf{x}}, \hat{\mathbf{z}}) \approx \min_{\theta} -\sum_{i=1}^{m} m_i \log p_\theta(x_i | \hat{\mathbf{x}}, \hat{\mathbf{z}}) - \sum_{j=1}^{n} n_j \log p_\theta(z_j | \hat{\mathbf{x}}, \hat{\mathbf{z}}), \tag{3}$$

where the notations are the same as in (1). Its fine-tuning is the same as that of BERT.

## 4 EXPERIMENTS

We make comparisons between AMBERT and the baselines including fine-grained BERT and coarse-grained BERT, as well as the alternatives including AMBERT-Combo and AMBERT-Hybrid,

using benchmark datasets in both Chinese and English. The experiments on the alternatives can also be seen as ablation study on AMBERT.

## 4.1 DATA FOR PRE-TRAINING

For Chinese, we use a corpus consisting of 25 million documents (57G uncompressed text) from Jinri Toutiao[1]. Note that there is no common corpus for training of Chinese BERT. For English, we use a corpus of 13.9 million documents (47G uncompressed text) from Wikipedia and OpenWeb-Text (Gokaslan & Cohen, 2019). Unfortunately, BookCorpus, one of the two corpora in the original paper for English BERT, is no longer publicly available.

The characters in the Chinese texts are naturally taken as fine-grained tokens. We conduct word segmentation on the texts and treat the words as coarse-grained tokens. We employ a word segmentation tool based on a n-gram model. Both tokenizations exploit WordPiece embeddings (Wu et al., 2016). There are 21,128 characters and 72,635 words in the vocabulary of Chinese.

The words in the English texts are naturally taken as fine-grained tokens. We perform coarse-grained tokenization on the English texts in the following way. Specifically, we first calculate the n-grams in the Wikipedia documents using KenLM (Heafield, 2011). We next build a phrase-level dictionary consisting of phrases whose frequencies are sufficiently high and whose last words highly depend on their previous words. We then employ a left-to-right search algorithm to perform phrase-level tokenization on the texts. There are 30,522 words and 77,645 phrases in the vocabulary of English.

## 4.2 EXPERIMENTAL SETUP

We make use of the same parameter settings for the AMBERT and BERT models. All models in this paper are 'base-models' having 12 layers of encoder. It is too computationally expensive for us to train the models as 'large models' having 24 layers. The hyper-parameters are basically the same as those in the original BERT paper (Devlin et al., 2018), which are given in Appendix C. The optimizer is Adam (Kingma & Ba, 2014). To enhance efficiency, we use mixed precision for all the models. Training is carried out on Nvidia V-100. The numbers of GPUs used for training are from 32 to 64, depending on the model sizes.

In pre-training of the AMBERT models, in total 15% of the coarse-grained tokens are masked, which is the same proportion for the BERT models. To retain consistency, the masked coarse-grained tokens are also masked as fine-grained tokens. In fine-tuning, we use the same hyper-parameters as those in the original papers of the baselines, and all the hyper-parameters are given in Appendix C.

## 4.3 CHINESE TASKS

### 4.3.1 BENCHMARKS

We use the benchmark datasets, Chinese Language Understanding Evaluation (CLUE) (Xu et al., 2020) for experiments in Chinese. CLUE contains six classification tasks, that are TNEWS, IFLYTEK and CLUEWSC2020, AFQMC, CSL and CMNLI[2], and three reading comprehension tasks which are CMRC2018, ChID and $C^3$. The details of all the benchmarks are shown in Appendix B. Data augmentation is also performed for all models in the tasks of TNEWS, CSL and CLUEWSC2020 to achieve better performances (see Appendix D for detailed explanation).

### 4.3.2 EXPERIMENTAL RESULTS

We compare AMBERT with the BERT baselines, including the BERT model released from Google, referred to as Google BERT, and the BERT model trained by us, referred to as Our BERT, including character based (fine-grained) and word based (coarse-grained) models. Case study in Appendix E.

Table 1 shows the results of the classification tasks. AMBERT improves average scores of the BERT baselines by about 1.0% and also works better than AMBERT-Combo and AMBERT-Hybrid. The results of Machine Reading Comprehensive (MRC) tasks are shown in Table 2. AMBERT improves average scores of the BERT baselines by over 3.0%. Our BERT (word) performs poorly in CMRC2018. This is probably because the results of word segmentation are not accurate enough for the task. AMBERT-Combo and AMBERT-Hybrid are on average better than single-grained BERT models. AMBERT further outperforms both of them.

---

[1] Jinri Toutiao is a popular news app. in China.
[2] The task is introduced at the CLUE website.

Table 1: Performances on classification tasks in CLUE in terms of accuracy (%). The numbers in boldface denote the best results of tasks. Average accuracies of models are also given. Numbers of parameters (param) and time complexities (cmplx) of models are also shown, where $l$, $n$, and $d$ denote layer number, sequence length, and hidden representation size respectively. The tasks with mark $\dagger$ are those with data augmentation.

| Model | Param. | Cmplx. | Avg. | TNEWS$^\dagger$ | IFLYTEK | CLUEWSC2020$^\dagger$ | AFQMC | CSL$^\dagger$ | CMNLI |
|---|---|---|---|---|---|---|---|---|---|
| Google BERT | 108M | $O(ln^2d)$ | 72.53 | 66.99 | **60.29** | 71.03 | 73.70 | 83.50 | 79.69 |
| Our BERT (char) | 108M | $O(ln^2d)$ | 71.90 | 67.48 | 57.50 | 70.69 | 71.80 | 83.83 | 80.08 |
| Our BERT (word) | 165M | $O(ln^2d)$ | 73.72 | 68.20 | 59.96 | 75.52 | 73.48 | 85.17 | 79.97 |
| AMBERT-Combo | 273M | $O(2ln^2d)$ | 73.61 | **69.60** | 58.73 | 71.03 | **75.63** | 85.07 | 81.58 |
| AMBERT-Hybrid | 176M | $O(4ln^2d)$ | 73.80 | 69.04 | 56.42 | 76.21 | 74.41 | 85.60 | 81.10 |
| AMBERT | 176M | $O(2ln^2d)$ | **74.67** | 68.58 | 59.73 | **78.28** | 73.87 | **85.70** | **81.87** |

Table 2: Performances on MRC tasks in CLUE in terms of F1, EM (Exact Match) and accuracy. The numbers in boldface denote the best results of tasks. Average scores of models are also given.

| Model | Avg. | CMRC2018 | | ChID | | $C^3$ | |
|---|---|---|---|---|---|---|---|
| | | DEV(F1,EM) | TEST(EM) | DEV(Acc.) | TEST(Acc.) | DEV(Acc.) | TEST(Acc.) |
| Google BERT | 73.76 | 85.48  64.77 | 71.60 | 82.20 | 82.04 | 65.70 | 64.50 |
| Our BERT (char) | 74.46 | 85.64  65.45 | 71.50 | 83.44 | 83.12 | 66.43 | 65.67 |
| Our BERT (word) | 65.77 | 81.87  41.69 | 41.30 | 80.89 | 80.93 | 66.72 | 66.96 |
| AMBERT-Combo | 75.26 | 86.12  65.11 | 72.00 | 84.53 | 84.64 | 67.74 | 66.70 |
| AMBERT-Hybrid | 75.53 | 86.71  68.16 | 72.45 | 83.37 | 82.85 | 67.45 | 67.75 |
| AMBERT | **77.47** | **87.29  68.78** | **73.25** | **87.20** | **86.62** | **69.52** | **69.63** |

We also compare AMBERT with the state-of-the-art models at the leader board of CLUE[3]. The base models, whose parameters are fewer than 200M, are trained with different datasets and procedures, and thus the comparisons should only be taken as references. Note that the settings of the base models are the same as that of Xu et al. (2020). Table 3 shows the results. The average score of AMBERT is higher than all the other models. We conclude that multi-grained tokenization is very helpful for pre-trained language models and the design of AMBERT is reasonable.

Table 3: State-of-the-art results of Chinese base models in CLUE.

| Model | Params | Avg. | TNEWS$^\dagger$ | IFLYTEK | WSC.$^\dagger$ | AFQMC | CSL$^\dagger$ | CMNLI | CMRC. | ChID | $C^3$ |
|---|---|---|---|---|---|---|---|---|---|---|---|
| Google BERT | 108M | 72.59 | 66.99 | 60.29 | 71.03 | 73.70 | 83.50 | 79.69 | 71.60 | 82.04 | 64.50 |
| XLNet-mid | 200M | 73.00 | 66.28 | 57.85 | 78.28 | 70.50 | 84.70 | 81.25 | 66.95 | 83.47 | 67.68 |
| ALBERT-xlarge | 60M | 73.05 | 66.00 | 59.50 | 69.31 | 69.96 | 84.40 | 81.13 | **76.30** | 80.57 | **70.32** |
| ERNIE | 108M | 74.20 | 68.15 | 58.96 | **80.00** | 73.83 | 85.50 | 80.29 | 74.70 | 82.28 | 64.10 |
| RoBERTa | 108M | 74.38 | 67.63 | **60.31** | 76.90 | **74.04** | 84.70 | 80.51 | 75.20 | 83.62 | 66.50 |
| AMBERT | 176M | **75.28** | **68.58** | 59.73 | 78.28 | 73.87 | **85.70** | **81.87** | 73.25 | **86.62** | 69.63 |

## 4.4 ENGLISH TASKS

### 4.4.1 BENCHMARKS

The General Language Understanding Evaluation (GLUE) benchmark (Wang et al., 2018) is a collection of nine NLU tasks. Following BERT (Devlin et al., 2018), we exclude the task WNLI for the reason that results of different models on this task are undifferentiated. In addition, three machine reading comprehensive tasks are also included, i.e., SQuAD v1.1, SQuAD v2.0, and RACE. The details of English benchmarks can be found in Appendix B.

### 4.4.2 EXPERIMENTAL RESULTS

We compare AMBERT with the BERT models on the tasks in GLUE. The results of Google BERT are from the original paper (Devlin et al., 2018), and the results of Our BERT are obtained by us. From Table 4 we can see the following trend. 1) Multi-grained models particularly AMBERT can achieve better results than single-grained models. 2) Among the multi-grained models, AMBERT performs best with fewer parameters and less computation. Case study is given in Appendix E.

We also make comparison on the SQuAD tasks. The results of Google BERT are either from the papers (Devlin et al., 2018; Yang et al., 2019) or from our runs with the official code. From Table 5 we make the following conclusions. 1) in SQuAD, AMBERT outperforms Google BERT with a large margin. Our BERT (word) generally performs well and Our BERT (phrase) performs poorly in the span detection tasks. 2) In RACE, AMBERT performs best among all the baselines for both development set and test set. 3) AMBERT is the best multi-grained model.

---

[3]The leader board of CLUE is at https://www.cluebenchmarks.com/rank.html.

Table 4: Performance on the tasks in GLUE. Average score over all the tasks is slightly different from the official GLUE score, since we exclude WNLI. CoLA uses Matthew's Corr. MRPC and QQP use both F1 and accuracy scores. STS-B computes Pearson-Spearman Corr. Accuracy scores are reported for the other tasks. Results of MNLI include MNLI-m and MNLI-mm. The other settings are the same as Table 1.

| Model | Param | Cmplx | Avg. | CoLA | SST-2 | MRPC | STS-B | QQP | MNLI | QNLI | RTE |
|---|---|---|---|---|---|---|---|---|---|---|---|
| Google BERT | 110M | $O(ln^2d)$ | 80.7 | 52.1 | 93.5 | 88.9/81.9 | 81.5/85.8 | 71.2/88.5 | 84.6/83.4 | 90.5 | 66.4 |
| Our BERT (word) | 110M | $O(ln^2d)$ | 81.6 | 53.7 | 93.8 | 88.8/84.8 | 84.3/86.0 | 71.6/89.0 | 85.0/84.5 | 91.2 | 66.8 |
| Our BERT (phrase) | 170M | $O(ln^2d)$ | 80.7 | 54.8 | 93.8 | 87.4/82.5 | 82.9/84.9 | 70.1/88.8 | 84.1/83.8 | 90.6 | 65.1 |
| AMBERT-Combo | 280M | $O(2ln^2d)$ | 81.8 | 57.1 | 94.5 | 89.2/84.8 | 84.4/85.8 | 71.8/88.6 | 84.7/84.2 | 90.4 | 66.2 |
| AMBERT-Hybrid | 194M | $O(4ln^2d)$ | 81.7 | 50.9 | 93.4 | 89.0/85.2 | 84.7/87.6 | 71.0/89.2 | 84.6/84.7 | 91.2 | 68.5 |
| AMBERT | 194M | $O(2ln^2d)$ | 82.7 | 54.3 | 94.5 | 89.7/86.1 | 84.7/87.1 | 72.5/89.4 | 86.3/85.3 | 91.5 | 70.5 |

Table 5: Performance on three English MRC tasks. We use EM and F1 to evaluate the performance of text detection, and report accuracies for RACE, on both development set and test set.

| Model | Avg. | SQuAD 1.1 | | SQuAD 2.0 | | | | RACE | |
|---|---|---|---|---|---|---|---|---|---|
| | | DEV(EM, F1) | | DEV(EM, F1) | | TEST(EM, F1) | | DEV | TEST |
| Google BERT | 74.0 | 80.8 | 88.5 | 70.1 | 73.5 | 73.7 | 76.3 | 64.5 | 64.3 |
| Our BERT (word) | 76.7 | 83.8 | 90.6 | 76.6 | 79.6 | 77.3 | 80.3 | 62.4 | 62.6 |
| Our BERT (phrase) | - | 67.4 | 82.3 | 55.4 | 62.6 | - | - | 66.9 | 66.1 |
| AMBERT-Combo | 77.2 | 84.0 | 90.9 | 76.4 | 79.6 | 76.6 | 79.8 | 66.6 | 63.7 |
| AMBERT-Hybrid | 77.3 | 83.6 | 90.3 | 76.4 | 79.4 | 76.7 | 79.7 | 67.1 | 65.1 |
| AMBERT | 78.6 | 84.2 | 90.8 | 77.6 | 80.6 | 78.6 | 81.4 | 68.9 | 66.8 |

We compare AMBERT with the state-of-the-art models in both GLUE[4] and MRC. The results of baselines, in Table 6, are either reported in published papers or re-implemented by us with HuggingFace's Transformer (Wolf et al., 2019). We use the provided implementation in HuggingFace's Transformer, without additional data augmentation or question-answering module[5]. Note that AMBERT outperforms all the models on average[6] without using training techniques such as bigger batches and dynamic masking.

Table 6: State-of-the-art results of English base models in GLUE. Each task only reports one score following Clark et al. (2020), and we report the average EM of SQuAD1.1 and SQuAD2.0 on development set. AMBERT[‡] is pre-trained with a corpora with size comparable to that of RoBERTa (160G uncompressed text). Scores with ⋆ are reported from the published papers.

| Model | Params | Avg. | CoLA | SST-2 | MRPC | STS-B | QQP | MNLI | QNLI | RTE | SQuAD | RACE |
|---|---|---|---|---|---|---|---|---|---|---|---|---|
| Google BERT | 110M | 78.7 | 52.1⋆ | 93.5⋆ | 84.8⋆ | 85.8⋆ | 89.2⋆ | 84.6⋆ | 90.5⋆ | 66.4⋆ | 75.5 | 64.3⋆ |
| XLNet | 110M | 78.6 | 47.9 | 94.3 | 83.3 | 84.1 | 89.2 | 86.8 | 91.7 | 61.9 | 79.9⋆ | 66.7⋆ |
| SpanBERT | 110M | 79.1 | 51.2 | 93.5 | 87.0 | 82.9 | 89.2 | 85.1 | 92.7 | 69.7 | 81.8 | 57.4 |
| ELECTRA | 110M | 81.3 | 59.7⋆ | 93.4⋆ | 86.7⋆ | 87.7⋆ | 89.1⋆ | 85.8⋆ | 92.7⋆ | 73.1⋆ | 74.8 | 69.9 |
| ALBERT | 12M | 80.1 | 53.2 | 93.2 | 87.5 | 87.2 | 87.8 | 85.0 | 91.2 | 71.1 | 78.7 | 65.8 |
| RoBERTa | 135M | 82.7 | 61.5 | 95.8 | 88.7 | 88.9 | 89.4 | 87.4 | 93.1 | 74.0 | 78.6 | 69.9 |
| AMBERT[‡] | 194M | 82.8 | 60.0 | 95.2 | 88.9 | 88.2 | 89.5 | 87.2 | 92.6 | 72.6 | 82.5 | 71.2 |

## 4.5 ENHANCEMENT OF INFERENCE SPEED

Although AMBERT can make significant improvements over single-grained models, the computation time is doubled. To enhance the inference speed of AMBERT, we develop a simplified version of it, referred to as AMBERT-Single, in which the two encoders are pre-trained and fine-tuned in learning and only the single-grained encoder is utilized in inference. Note that it has the same amount of inference computation as BERT. We conduct experiments on CLUE/GLUE/SQuADs/RACE with AMBERT-Single. The results on the development sets are shown in Table 7. We conclude that, a) for the English tasks, AMBERT-Single achieves similar results as AMBERT and outperforms "Our BERT (Single)" with a large margin using the same amount of inference time; b) for the Chinese tasks, AMBERT-Single is slightly worse than AMBERT and performs much better than "Our BERT (Single)". Therefore, in practice, one can train an AMBERT with two encoders and use only one of them in inference, i.e., AMBERT-Single.

---

[4]The leader board of GLUE is at https://gluebenchmark.com/leaderboard.

[5]For that reason, we cannot use the results for SQuAD 2.0 in Clark et al. (2020).

[6]In the previous versions, we reported the results (average score 82.3) of AMBERT when we were only able to use a smaller dataset for pre-training in English.

Table 7: Performances on the development sets of CLUE, GLUE, SQuAD and RACE with AMBERT-Single or Our BERT (better one) for inference. CN-Models and EN-Models denote Chinese and English pre-trained models respectively. CoLA uses Matthew's Corr. We report EM of CMRC2018 and the average EM of SQuAD1.1 and SQuAD2.0. The other metrics are all accuracies.

| CN-Models | Speedup | Avg. | TNEWS | IFLYTEK | CLUEWSC2020 | AFQMC | CSL | CMNLI | CMRC2018 | ChID | $C^3$ | - |
|---|---|---|---|---|---|---|---|---|---|---|---|---|
| AMBERT | 1.0 | 75.3 | 68.1 | 60.1 | 81.6 | 74.7 | 85.6 | 82.3 | 68.8 | 87.1 | 69.2 | - |
| AMBERT-Single | 2.0x | 74.8 | 68.0 | 59.5 | 81.3 | 74.2 | 85.5 | 82.1 | 67.4 | 86.6 | 68.5 | - |
| Our BERT (Single) | 2.0x | 73.4 | 67.8 | 58.7 | 79.0 | 74.1 | 84.5 | 80.8 | 65.5 | 83.4 | 66.7 | - |
| EN-Models | Speedup | Avg. | CoLA | SST-2 | MRPC | STS-B | QQP | MNLI | QNLI | RTE | SQuAD | RACE |
| AMBERT | 1.0 | 79.0 | 61.7 | 94.3 | 92.3 | 55.0 | 91.2 | 86.2 | 91.3 | 70.2 | 80.9 | 66.8 |
| AMBERT-Single | 2.0x | 78.9 | 62.2 | 93.2 | 92.5 | 55.0 | 91.2 | 86.1 | 91.4 | 70.6 | 80.3 | 66.7 |
| Our BERT (Single) | 2.0x | 77.1 | 56.6 | 92.4 | 89.7 | 54.2 | 90.4 | 85.1 | 90.6 | 69.1 | 80.2 | 62.6 |

## 4.6 REGULARIZATION IN FINE-TUNING

Table 8 shows the results of using different values as regularization coefficients in fine-tuning on the development sets of CLUE, GLUE and RACE. It appears that for most tasks the use of regularization is necessary. For simplicity, we did not use the best value of coefficient for each task and instead we adopt 0.0 for RACE and 1.0 for the other tasks.

Table 8: Performances on the development sets of CLUE, GLUE and RACE with different regularization coefficients in fine-tuning. CN-Models and EN-Models stand for Chinese and English pre-trained models respectively. CoLA uses Matthew's Corr. The other metrics are accuracies.

| CN-Models | $\lambda$ | TNEWS | IFLYTEK | CLUEWSC2020 | AFQMC | CSL | CMNLI | ChID | $C^3$ | - |
|---|---|---|---|---|---|---|---|---|---|---|
| AMBERT | 1.0 | **68.1** | 60.1 | **81.6** | 74.7 | **85.6** | **82.3** | 87.1 | 69.2 | - |
| AMBERT | 0.0 | 67.9 | **60.3** | 80.9 | **75.4** | 85.0 | 81.1 | 86.5 | 69.2 | - |
| EN-Models | $\lambda$ | CoLA | SST-2 | MRPC | STS-B | QQP | MNLI | QNLI | RTE | RACE |
| AMBERT | 1.0 | **61.7** | **94.3** | **92.3** | **55.0** | **91.2** | **86.2** | **91.3** | 70.2 | 66.6 |
| AMBERT | 0.0 | 61.5 | 93.4 | 90.1 | 54.5 | 91.1 | 85.5 | 91.2 | **70.2** | **66.8** |

## 4.7 DISCUSSIONS

We further investigate the reason that AMBERT is superior to AMBERT-Combo. Figure 2 shows the distances between the [CLS] representations of the fine-grained encoder and coarse-grained encoder in AMBERT-Combo and AMBERT after pre-training, in terms of cosine dissimilarity (one minus cosine similarity) and normalized Euclidean distance. One can see that the distances in AMBERT-Combo are larger than the distances in AMBERT in the tasks. We perform the assessment using the data in the other tasks and find similar trends. The results indicate that the representations of fine-grained encoder and coarse-grained encoder are closer in AMBERT than in AMBERT-Combo. These are natural consequences of using AMBERT and AMBERT-Combo, whose parameters are respectively shared and unshared across encoders. It implies that the higher performances by AMBERT is due to its parameter sharing, which can use less parameters to learn and represent similar ways of combining tokens no matter whether they are fine-grained or coarse-grained. An intuitive explanation is that the ways of combining representations of fine-grained tokens and the ways of combining representations of coarse-grained tokens "in the same contexts" are exactly the same.

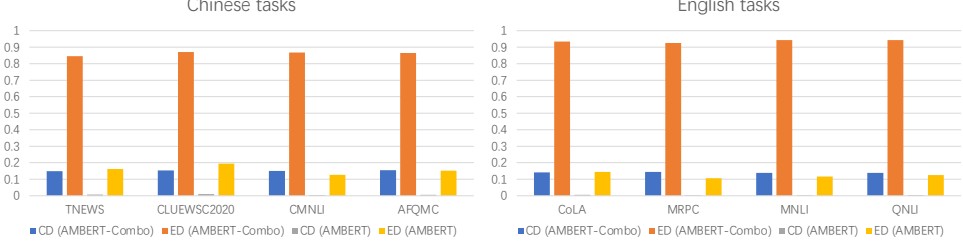

Figure 2: Distances between representations of fine-grained and coarse-grained encoders (representations of [CLS]) in AMBERT-Combo and AMBERT. CD and ED stand for cosine dissimilarity (one minus cosine similarity) and normalized Euclidean distance respectively.

We also examine the reasons that AMBERT works better than AMBERT-Hybrid, while both of them exploit multi-grained tokenization. Figure 3 shows the attention weights of first layers in AMBERT and AMBERT-Hybrid, as well as the single-grained BERT models, after pre-training. In AMBERT-Hybrid, the fine-grained tokens attend more to the corresponding coarse-grained tokens and as a

result the attention weights among fine-grained tokens are weakened. In contrast, in AMBERT the attention weights among fine-grained tokens and those among coarse-grained tokens are intact. It appears that attentions among single-grained tokens (fine-grained ones and coarse-grained ones) play important roles in downstream tasks.

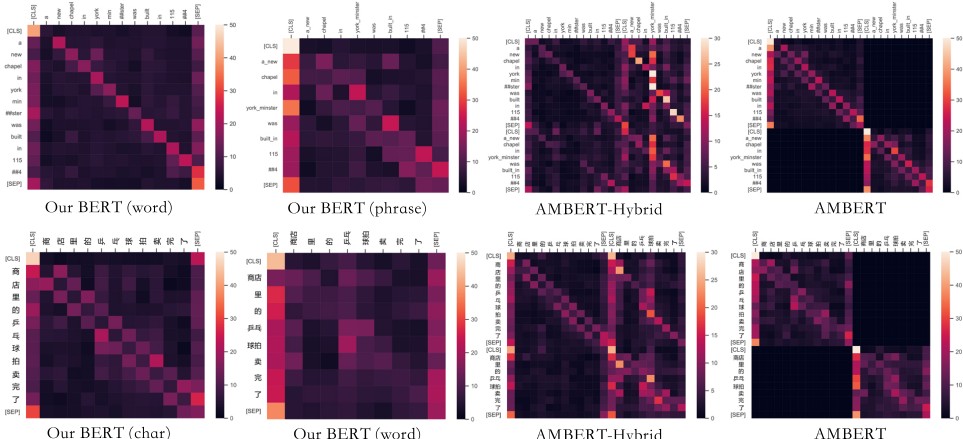

Figure 3: Attention weights of first layers of Our BERT (word/phrase), AMBERT-Hybrid and AMBERT, for English and Chinese sentences.

To answer the question why the improvements by AMBERT on Chinese are larger than on English in the same pre-training settings, we further make an analysis. We tokenize 10,000 randomly selected Chinese sentences in CLUE with our Chinese word tokenizer. As shown in Table 9, the average proportion of words is 51.5%, which indicates that about half of the tokens are fine-grained and half are coarse-grained in Chinese. We also tokenize 10,000 randomly selected English sentences in GLUE with our English phrase tokenizer. The average proportion of phrases is only 13.1%, which means that there are much less coarse-grained tokens than fine-grained tokens in English. Therefore, we postulate that for Chinese it is necessary for a model to process the language at both fine-grained and coarse-grained levels. AMBERT indeed has the capability.

Table 9: The rate of coarse-grained tokens (those not included in fine-grained vocabulary) in coarse-grained tokenization.

| Datasets | Chinese words | Chinese total tokens | word rate (%) | English phrases | English total tokens | phrase rate (%) |
|---|---|---|---|---|---|---|
| CMNLI/MNLI | 157,511 | 335,187 | 47.0 | 43,661 | 318,985 | 13.7 |
| TNEWS/QNLI | 71,636 | 137,965 | 51.9 | 59,506 | 395,681 | 15.0 |
| TNEWS/QNLI | 94,439 | 165,847 | 56.9 | 35,256 | 237,731 | 14.8 |
| CSL/SST-2 | 836,976 | 1,739,954 | 48.1 | 9,757 | 103,048 | 9.47 |
| CHID/CoLA | 958,893 | 1,763,507 | 53.4 | 10,491 | 82,353 | 12.7 |
| Avg. | - | - | 51.5 | - | - | 13.1 |

## 5 CONCLUSION

In this paper, we have proposed a novel pre-trained language model called AMBERT, as an extension of BERT. AMBERT employs multi-grained tokenization, that is, it uses both words and phrases in English and both characters and words in Chinese. With multi-grained tokenization, AMBERT learns in parallel the representations of the fine-grained tokens and the coarse-grained tokens using two encoders with shared parameters. Experimental results have demonstrated that AMBERT significantly outperforms BERT and other models in NLU tasks in both English and Chinese. AMBERT increases average score of Google BERT by about 2.7% in Chinese benchmark CLUE. AMBERT improves Google BERT by over 3.0% on a variety of tasks in English benchmarks GLUE, SQuAD (1.1 and 2.0), and RACE. We also develop AMBERT-Simple which performs equally well as AMBERT with about half of inference time.

As future work, we plan to study the following issues: 1) to investigate model acceleration methods in learning of AMBERT, such as sparse attention (Child et al., 2019; Kitaev et al., 2020; Zaheer et al., 2020) and synthetic attention (Tay et al., 2020); 2) to apply the technique of AMBERT into other pre-trained language models such as XLNet; 3) to employ AMBERT in other NLU tasks.

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

# A  ATTENTION MAPS FOR SINGLE-GRAINED MODELS

We construct fine-grained and coarse-grained BERT models for English and Chinese, and examine the attention maps of the models using the BertViz tool (Vig, 2019). Figure A shows the attention maps of the first layer of fine-grained models for several sentences in English and Chinese. One can see that there are tokens that improperly attend to other tokens in the sentences. For example, in the English sentences, the words "drawing", "new", and "dog" have high attention weights to "portrait", "york", and "food", respectively, which are not appropriate. For example, in the Chinese sentences, the chars "拍", "北", "长" have high attention weights to "卖", "京", "市", respectively, which are also not reasonable. (It is verified that the bottom layers at BERT mainly represent lexical information, the middle layers mainly represent syntactic information, and the top layers mainly represent semantic information (Jawahar et al., 2019).) Ideally a token should only attend to the tokens with which they form a lexical unit at the first layer. This cannot be guaranteed in the fine-grained BERT model, however, because usually a fine-grained token may belong to multiple lexical units (i.e., there is ambiguity).

Figure 5 shows the attention maps of the first layer of coarse-grained models for the same sentences in English and Chinese. In the English sentences, the words are combined into the phrases of "drawing room", "york minister", and "dog food". The attentions are appropriate in the first two sentences, but it is not in the last sentence because of the incorrect tokenization. Similarly, in the Chinese sentences, the high attention weights of words " 球拍(bat)" and "京城(capital)" are reasonable, but that of word "市长(mayor)" is not. Note that incorrect tokenization is inevitable.

# B  DETAILED DESCRIPTIONS FOR THE BENCHMARKS

## B.1  CHINESE TASKS

TNEWS is a text classification task in which titles of news articles in TouTiao are to be classified into 15 classes. IFLYTEK is a task of assigning app descriptions into 119 categories. CLUEWSC2020, standing for the Chinese Winograd Schema Challenge, is a co-reference resolution task. AFQMC is a binary classification task that aims to predict whether two sentences are semantically similar. CSL uses the Chinese Scientific Literature dataset containing abstracts and their keywords of papers and the goal is to identify whether given keywords are the original keywords of a paper. CMNLI is based on translation from MNLI (Williams et al., 2017), which is a large-scale, crowd-sourced entailment classification task. CMRC2018 (Cui et al., 2018) makes use of a span-based dataset for Chinese machine reading comprehension. ChID (Zheng et al., 2019) is a large-scale Chinese IDiom cloze test. $C^3$ (Sun et al., 2019a) is a free-form multiple-choice machine reading comprehension for Chinese.

## B.2  ENGLISH TASKS

CoLA (Warstadt et al., 2019) contains English acceptability judgments drawn from books and journal articles on linguistic theory. SST-2 (Socher et al., 2013) consists of sentences from movie reviews

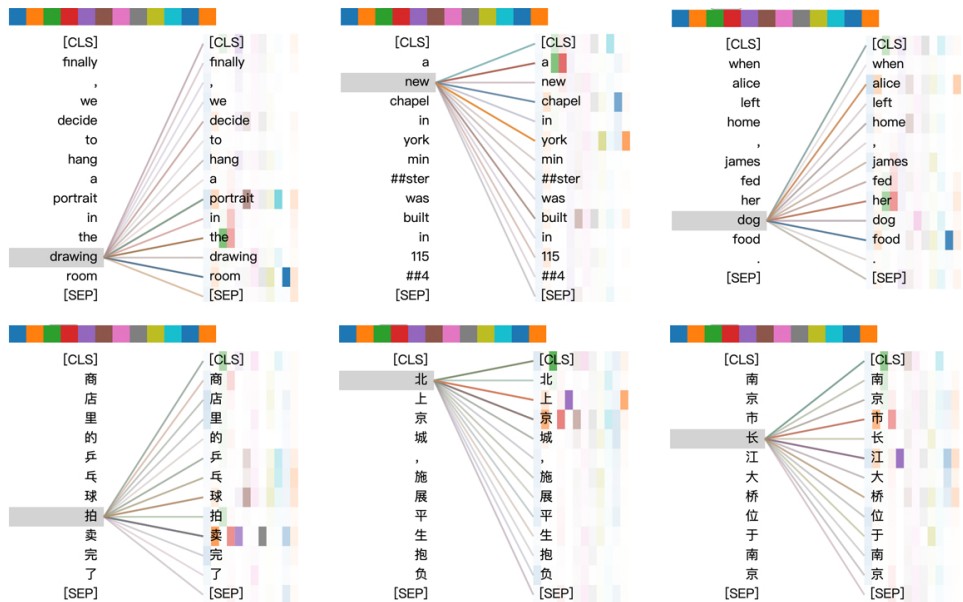

Figure 4: Attention maps of first layers of fine-grained BERT models for English and Chinese sentences. The Chinese sentences are "商店里的兵乓球拍卖完了 (Table tennis bats are sold out in the shop)", "北上京城施展平生报复 (Go north to Beijing to fulfill the dream)", "南京市长江大桥位于南京 (The Nanjing Yantze River bridge is located in Nanjing)". Different colors represent attention weights in different heads and darkness represents weight.

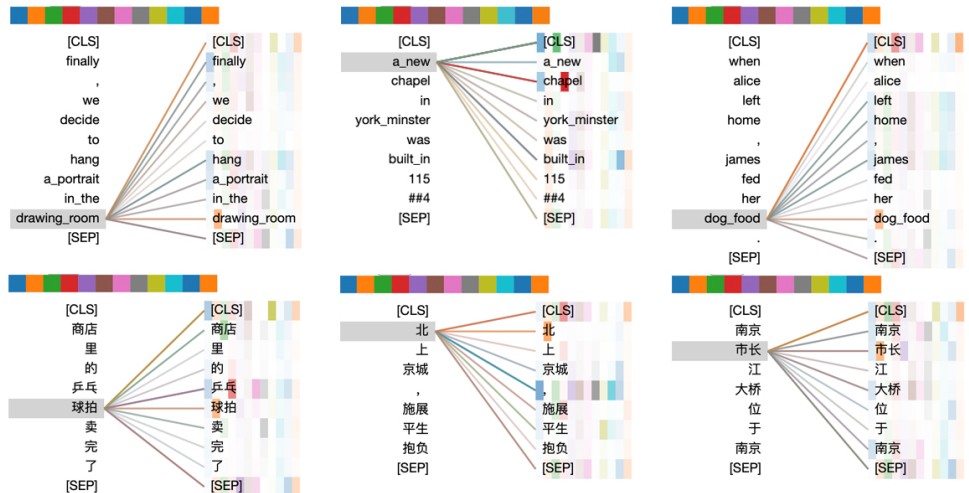

Figure 5: Attention maps of first layers of coarse-grained BERT models for English and Chinese sentences. Note that tokenizations may have errors.

and human annotations of their sentiment. MRPC (Dolan & Brockett, 2005) is a corpus of sentence pairs automatically extracted from online news sources, and the target is to identify whether a sentence pair is semantically equivalent. STS-B (Cer et al., 2017) is a collection of sentence pairs and the task is to predict similarity scores. QQP is a collection of question pairs and requires models to recognize semantically equivalent ones. MNLI (Williams et al., 2017) is a crowd-sourced collection of sentence pairs with textual entailment annotations. QNLI (Wang et al., 2018) is a question-answering dataset consisting of question-paragraph pairs, where one of the sentences in the paragraph contains the answer to the corresponding question. RTE (Bentivogli et al., 2009) comes from a series of annual textual entailment challenges.

## C    HYPER-PARAMETERS

### C.1    HYPER-PARAMETERS IN PRE-TRAINING

We adopt the standard hyper-parameters of BERT in pre-training of the models except batch sizes which are tuned to make our fine-grained BERT models comparable to the Google BERT models. Table 10 shows the hyper-parameters in our Chinese AMBERT and English AMBERT. Our BERT models and alternatives of AMBERT (AMBERT-Combo and AMBERT-Hybrid) all use the same hyper-parameters in pre-training.

Table 10: Hyper-parameters for pre-trained AMBERT.

| Hyperparam | Chinese AMBERT | English AMBERT |
|---|---|---|
| Number of Layers $l$ | 12 | 12 |
| Hidden Size $d$ | 768 | 768 |
| Sequence Lengh $n$ | 512 | 512 |
| FFN Inner Hidden Size | 3072 | 3072 |
| Attention Heads | 12 | 12 |
| Attention Head Size | 64 | 64 |
| Dropout | 0.1 | 0.1 |
| Attention Dropout | 0.1 | 0.1 |
| Warmup Steps | 10,000 | 10,000 |
| Peak Learning Rate | 1e-4 | 1e-4 |
| Batch Size | 512 | 1024 |
| Weight Decay | 0.01 | 0.01 |
| Max Steps | 1m | 500k |
| Learning Rate Decay | Linear | Linear |
| Adam $\epsilon$ | 1e-6 | 1e-6 |
| Adam $\beta_1$ | 0.9 | 0.9 |
| Adam $\beta_2$ | 0.999 | 0.999 |

### C.2    HYPER-PARAMETERS IN FINE-TUNING

For the Chinese tasks, since all the original papers do not report detailed hyper-parameters in fine-tuning of the baseline models, we uniformly use the same hyper-parameters as shown in Table 11 except training epoch, because AMBERT and AMBERT-Combo have more parameters and need more training to get converged. We choose the training epochs for all models when the performances on development sets stop to improve. As for the English tasks, Table 12 shows all the hyper-parameters in fine-tuning of the models. We adopt the best hyper-parameters in the original papers for the baselines. Moreover, for AMBERT[‡], we also tune learning rate ([1e-5, 2e-5, 3e-5]) and batch size ([16, 32]) for GLUE with the same method in RoBERTa (Liu et al., 2019).

Table 11: Hyper-parameters for fine-tuning of Chinese tasks.

| Dataset | Modes | Batch Size | Max Length | Epoch | Learning Rate | $\lambda$ |
|---|---|---|---|---|---|---|
| TNEWS/IFLYTEK/AFQMC/CSL/CMNLI | Our BERT (char) | 32 | 128 | 5 | 2e-5 | - |
| | Our BERT (word) | 32 | 128 | 5 | 2e-5 | - |
| | AMBERT-Combo | 32 | 128 | 8 | 2e-5 | 1.0 |
| | AMBERT-Hybrid | 32 | 128 | 5 | 2e-5 | - |
| | AMBERT | 32 | 128 | 8 | 2e-5 | 1.0 |
| CLUEWSC2020 | Our BERT (char) | 8 | 128 | 50 | 2e-5 | - |
| | Our BERT (word) | 8 | 128 | 50 | 2e-5 | - |
| | AMBERT-Combo | 8 | 128 | 80 | 2e-5 | 1.0 |
| | AMBERT-Hybrid | 8 | 128 | 50 | 2e-5 | - |
| | AMBERT | 8 | 128 | 80 | 2e-5 | 1.0 |
| CMRC2018 | All the models | 32 | 512 | 2 | 2e-5 | - |
| ChID | Our BERT, AMBERT-Hybrid | 24 | 64 | 3 | 2e-5 | - |
| | AMBERT, AMBERT-Combo | 24 | 64 | 3 | 2e-5 | 1.0 |
| $C^3$ | Our BERT, AMBERT-Hybrid | 24 | 512 | 8 | 2e-5 | - |
| | AMBERT, AMBERT-Combo | 24 | 512 | 8 | 2e-5 | 1.0 |

## D    DATA AUGMENTATION

To enhance the performance, we conduct data augmentation for the three Chinese classification tasks of TNEWS, CSL, and CLUEWSC2020. In TNEWS, we use both keywords and titles. In CSL, we concatenate keywords with a special token "_". In CLUEWSC2020, we duplicate a few instances having pronouns in the training data such as "她 (she)".

Table 12: Hyper-parameters for fine-tuning of English tasks.

| Dataset | Modes | Batch Size | Max Length | Epoch | Learning Rate | $\lambda$ |
|---|---|---|---|---|---|---|
| SST-2/MRPC/QQP/MNLI/QNLI | Our BERT (word) | 32 | 512 | 4 | 2e-5 | - |
| | Our BERT (phrase) | 32 | 512 | 4 | 2e-5 | - |
| | AMBERT-Combo | 32 | 512 | 6 | 2e-5 | 1.0 |
| | AMBERT-Hybrid | 32 | 512 | 4 | 2e-5 | - |
| | AMBERT | 32 | 512 | 6 | 2e-5 | 1.0 |
| CoLA/STS-B | Our BERT (word) | 32 | 512 | 10 | 2e-5 | - |
| | Our BERT (phrase) | 32 | 512 | 10 | 2e-5 | - |
| | AMBERT-Combo | 32 | 512 | 20 | 2e-5 | 1.0 |
| | AMBERT-Hybrid | 32 | 512 | 10 | 2e-5 | - |
| | AMBERT | 32 | 512 | 20 | 2e-5 | 1.0 |
| RTE | Our BERT (word) | 32 | 512 | 20 | 2e-5 | - |
| | Our BERT (phrase) | 32 | 512 | 20 | 2e-5 | - |
| | AMBERT-Combo | 32 | 512 | 50 | 2e-5 | 1.0 |
| | AMBERT-Hybrid | 32 | 512 | 20 | 2e-5 | - |
| | AMBERT | 32 | 512 | 50 | 2e-5 | 1.0 |
| SQuAD (1.1 and 2.0) | All the models | 32 | 512 | 3 | 2e-5 | - |
| RACE | All except the following two | 16 | 512 | 4 | 1e-5 | - |
| | AMBERT-Combo | 32 | 512 | 6 | 1e-5 | 0.0 |
| | AMBERT | 32 | 512 | 6 | 1e-5 | 0.0 |

# E CASE STUDY

Table 13: Case study for sentence matching tasks in both English and Chinese (QNLI and CMNLI). The value "0" denotes entailment relation, while the value "1" denotes no entailment relation. WORD/PHRASE represents Our BERT word/phrase. In English the tokens in the same phrase are concatenated with "_", and in Chinese phrases are split with "/".

| Sentence1 | Sentence2 | Label | WORD | PHRASE | AMBERT |
|---|---|---|---|---|---|
| What Star Trek episode has a nod to Doctor Who? (What Star_Trek episode has_a nod to Doctor_Who?) | There have also been many references to Doctor Who in popular culture and other science fiction, including Star Trek: The Next Generation ("The Neutral Zone") and Leverage. (There have also been many references_to Doctor_Who in popular_culture and_other science_fiction, including Star_Trek: the_next_generation ("the_neutral_zone") and leverage.) | 0 | 1 | 0 | 0 |
| What was the name of the blind date concept program debuted by ABC in 1966? (What was_the name_of the_blind date concept program debuted by ABC in_1966?) | In December of that year, the ABC television network premiered The Dating Game, a pioneer series in its genre, which was a reworking of the blind date concept in which a suitor selected one of three contestants sight unseen based on the answers to selected questions. (In_December of that_year, the_ABC television_network premiered the dating game, a pioneer series in_its genre, which was_a reworking of_the blind_date concept in_which a suitor selected one_of_three contestants sight unseen based_on_the answers to selected questions.) | 0 | 0 | 1 | 0 |
| What are two basic primary resources used to guage complexity? (What are two basic primary resources used_to guage complexity?) | The theory formalizes this intuition, by introducing mathematical models of computation to study these problems and quantifying the amount of resources needed to solve them, such as time and storage. (The theory formalizes this intuition, by introducing mathematical models of computation to study these problems and quantifying the_amount_of resources needed to_solve them, such_as time and storage.) | 0 | 1 | 1 | 0 |
| What is the frequency of the radio station WBT in North Carolina? (What_is the_frequency_of the_radio_station WBT in_north_carolina?) | WBT will also simulcast the game on its sister station WBTFM (99.3 FM), which is based in Chester, South Carolina. (WBT will also simulcast the_game on_its sister_station WBTFM (99.3 FM), which_is based_in Chester, South_Carolina.) | 1 | 0 | 0 | 1 |
| 只打那些面对我们的人，乔恩告诉阿德林。 (只/打/那些/面对/我们/的/人，/乔恩/告诉/阿/德/林/。) | "打死那些面对我们的人，" 阿德林对乔恩说。 ("/打死/那些/面对/我们/的/人，/"/阿/德/林/对/乔恩/说/。。) | 1 | 0 | 1 | 1 |
| 教堂有一个更精致的巴洛克讲坛。 (教堂/有/一个/更/精致/的/巴洛克/讲坛/。) | 教堂有一个巴罗格式的讲坛。 (教堂/有/一个/巴/罗/格式/的/讲坛/。) | 0 | 0 | 1 | 0 |
| 我们已经采取了一系列措施来增强我们员工的能力，并对他们进行投资。 (我/们/已 经/采 取/了/一/系 列/措 施/来/增 强/我/们/员 工/的/能 力，/并/对/他们/进行/投资/。) | 我们一定会投资在我们的工人身上。 (我们/一定/会/投资/在/我们/的/工人/身上/。) | 0 | 1 | 1 | 0 |
| 科技行业的故事之所以活跃起来，是因为现实太平淡了。 (科 技/行 业的/故 事/之 所 以/活 跃/起 来/，/是/因 为/现 实/太 平/淡/了/。) | 现实是如此平淡，以致于虚拟现实技术业务得到了刺激。 (现实/是/如此/平淡/，/以致/于/虚拟/现实/技术/业务/得到/了/刺激/。。) | 1 | 0 | 0 | 1 |

We also qualitatively study the results of BERT and AMBERT, and find that they support our claims (cf., Section 1) very well. Here, we give some random examples from the entailment tasks (QNLI and CMNLI) in Table 13. One can have the following observations. 1) The fine-grained models (e.g., Our BERT word) cannot effectively use complete lexical units such as "Doctor Who" and "打死" (sentence pairs 1 and 5), which may result in incorrect predictions. 2) The coarse-grained models (e.g., Our BERT phrase), on the other hand, cannot effectively deal with incorrect tokenizations, for

example, "the blind" and "格式" (sentence pairs 2 and 6). 3) AMBERT is able to make effective use of complete lexical units such as "sister station" in sentence pair 4 and "员工/ 工人" in sentence pair 7, and robust to incorrect tokenizations, such as "used to" in sentence pair 3. 4) AMBERT can in general make more accurate decisions on difficult sentence pairs with both fine-grained and coarse-grained tokenization results.

