# OpenReview forum: "AMBERT: A Pre-trained Language Model with Multi-Grained Tokenization"
_ICLR.cc/2021/Conference — Reject_

### Official Review · AnonReviewer3 · 2020-10-27
**Initial Review**

**Rating:** 5
**Confidence:** 4

**Review:**

[General Review]
In this paper, the authors propose a new pre-trained language model called AMBERT, which focuses on both fine-grained and coarse-grained tokenizations.
The idea of the model is straightforward.
The proposed AMBERT adopts two encoders:
1) fine-grained: for encoding char-level (Chinese) or sub-word level (English) information.
2) coarse-grained: for encoding n-gram information.
The final hidden representations of two encoders are concatenated and perform MLM+NSP pre-training tasks.
The authors also propose two alternatives AMBERT-Combo (without parameter sharing between two encoders) and AMBERT-Hybrid (using one encoder for both granularities).

They carried out experiments on both Chinese (CLUE) and English (GLUE, SQuAD 1.1, SQuAD 2.0, RACE) benchmarks.
The experimental results show the proposed AMBERT could achieve significant improvements over various baseline systems.

Overall, the design of the model is straightforward, and the paper is easy to read. However, I also have several concerns about this paper.
1) Using multi-granularities in a pre-trained language model is not novel, considering the SpanBERT (Joshi et al., 2020) and ZEN (Diao et al., 2019) has already been existing for some time.
2) The source of the improvements is not clear. As the proposed model uses about 2x parameters of BERT-base, it is not clear whether the improvements are benefited from these additional parameters or the design of the multi-grained encoders. Note that they only perform the experiments on base-level models but not on large-level models, which indicates that the pre-trained models could still benefit a lot from increasing the parameter sizes.

Based on these observations, I am leaning towards a weak rejection of the paper (but I'm open to discussing).


[Strengths]
1. Demonstrate that the combination of word-level and phrase-level information is helpful in pre-trained language models.
2. Good experimental results on the existing Chinese and English benchmarks.

[Weaknesses]
1. The technical novelty is limited. There is nothing much exciting to know that the model could achieve better scores, considering its parameter size (about 2x of BERT-base) and the use of variants of whole word masking or n-gram masking.
2. Some of the implementation details are missing, and the replication of the results remains uncertain (no supplementary codes or pseudo implementation provided).


[Questions for Authors]
1. I've checked Appendix C.1, but did not find if your model is pre-trained from scratch or starting from BERT checkpoint by Devlin et al. (2018)? I think the fine-grained encoder is the same with original Chinese BERT.
2. For the Chinese word segmentation, the authors mention that 'a word segmentation tool based on an n-gram model'. I am not sure if the CWS tool is in-house implemented? If so, how was the word vocabulary (72,635) obtained? Was it a truncated list by frequency order?
3. The proposed AMBERT on Chinese MRC got worse results on CMRC and C^3 on CLUE (Table 3). The authors indicate that this may be caused by the inaccurate CWS for these tasks. However, when it comes to English counterparts, AMBERT shows similar or better performance than state-of-the-art counterparts. I am quite confused about these results. Could you explain more about this? (maybe it is also related to Question 2)


[Minor Comments]
1. page 2, section 1: as extension -> as an extension
2. page 4, section 3.2: (Next sentence prediction -> Next sentence prediction
3. page 5, section 4.3.1: reading-comprehension -> reading comprehension
4. It is better to mark as '[CLS]' instead of 'CLS' in Figure 1. Also for '[SEP]'.

---

> ### Author Response · Authors · 2020-11-23
> **Response to AnonReviewer3**
>
> Thank you for your review.
>
> For your concerns:
> 1. Using multi-granularities in a pre-trained language model is not novel, considering the SpanBERT (Joshi et al., 2020) and ZEN (Diao et al., 2019) has already been existing for some time.
>
> We made discussions on SpanBERT and ZEN in Section 2 and other coarse-grained tokenization related techniques such as ERINE and "whole word masking". The differences between AMBERT and the existing works are significant. The key question is not whether or not to use coarse-grained tokens, but is how to do it.
>
> 2. The source of the improvements is not clear.
>
> Note that the number of parameters does not increase so much, compared with BERT (less than a 15% increase compared with the coarse-grained model), because the parameters in the two encoders in AMBERT are shared. There are only additional parameters from multi-grained embeddings. The increased parameters in Large models are layers, heads, and hidden states, which are orthogonal to the approach we are taking. (Due to limitation of available computation resources we are not able to train Large models.)
>
> The improvements over the existing models are from the architecture of AMBERT, as demonstrated in the experiments. Results in table [1, 2, 4, 5] show that AMBERT is much better than single-grained models.
>
> For the weakness:
> 1. The technical novelty is limited.
>
> Our work on AMBER is more like a discovery than an invention. We find that with the architecture one can achieve better performances than the existing models. This cannot be easily found and was not known so far. There are also comprehensive experiments on the comparisons. All of these should be valuable for future research. We think, therefore, that our work represents a significant contribution.
>
> 2. Some of the implementation details are missing.
>
> Due to our company's policy, we cannot release the code and the model. We will look for partners to help re-implement the tool and make it publicly available.
>
> For the questions:
> 1. I've checked Appendix C.1, but did not find if your model is pre-trained from scratch or starting from BERT checkpoint by Devlin et al.(2018)? I think the fine-grained encoder is the same as the original Chinese BERT.
>
> We did not use the BERT checkpoint by Devlin et al.(2018). All the models were trained with random initialization. The fine-grained encoder is indeed the same as the original Chinese BERT.
>
> 2. For the Chinese word segmentation, the authors mention that'a word segmentation tool based on an n-gram model. I am not sure if the CWS tool is in-house implemented? If so, how was the word vocabulary (72,635) obtained? Was it a truncated list by frequency order?
>
> The segmentation tool is in-house implemented with standard techniques such as those in Jieba. We built the vocabulary depending on word frequencies. In Chinese, 'Our BERT (word)' and AMBERT used the same segmentation tool.
>
> 3. The proposed AMBERT on Chinese MRC got worse results on CMRC and C^3 on CLUE (Table 3). The authors indicate that this may be caused by the inaccurate CWS for these tasks. However, when it comes to English counterparts, AMBERT shows similar or better performance than state-of-the-art counterparts. I am quite confused about these results. Could you explain more about this? (maybe it is also related to Question 2)
>
> The base models in Table 3 (Chinese) and Table 6 (English) are trained with different datasets and procedures, and thus their results should only be taken as references. It happens that in the tables AMBERT appears to work better in English than in Chinese for MRC. In contrast, in Table 2 (Chinese) and Table 4 (English), AMBERT is compared fairly to single grained models under the same experimental settings. Thus we can conclude that AMBERT can consistently outperform single-grained models on MRC tasks. Note that we only reported that 'Our BERT (word)', not AMBERT, works poorly on CMRC and we conjecture that it is due to Chinese word segmentation.

---

### Official Review · AnonReviewer1 · 2020-10-27
**Experiments not Comprehensive**

**Rating:** 3
**Confidence:** 4

**Review:**

This paper proposed to use both fine and coarse grained tokenizations of text to train large language models like BERT. The method is relatively straightforward. The input is tokenized at different two granularities (words and phrases for English; characters and words for Chinese). Each type of tokenized text is passed through BERT layers with shared parameters to generate contextual representations. At the finetuning stage, both fine grained and coarse grained representations of the CLS token are jointly used to predict the target.

I was not sure about what exactly was Our Bert (word). Is it not using the BERT WordPiece tokenizer ? If not, which tokenizer is it using ? Also, if it is not using BERT tokenizer, I am very surprised to see it perform better than Google BERT in all cases. Since Google BERT uses a more fine grained tokenization, I would expect it to perform equally well or better. Please clarify this part.

There are two baselines which I think are important for evaluation. First is a BPE tokenizer trained on the data you are using. This might resolve the problems with single granularity for both language considered here. Second is the "whole-word masking" approach of BERT. You can use fine grained tokenization but mask out whole words / phrases as the case may be, which might give you the best of both worlds. Without these baselines, I am not convinced that we should prefer Ambert to other approaches.

---

> ### Author Response · Authors · 2020-11-23
> **Response to AnonReviewer1**
>
> Thank you for your review.
>
> Yes, all of 'Our BERT' models used the Google BERT WordPiece tokenizer. Not all the data for pre-training of Google BERT is publicly available. We did our best to reproduce Google BERT and used a different dataset for pre-training to obtain Our BERT, as explained. Note that the variances of the results of Our BERT and Google BERT are small. Our BERT does not always work better than Google BERT. For example, in RACE in English, Our BERT works slightly worse than Google BERT. Please see Tables 1, 2, 4, and 5. The key point is that we used exactly the same settings for comparison between AMBERT and the '`Our BERT' baselines.
>
> The key methods in the two baselines which you are mentioning are 'Google BPE tokenization' and 'whole-word masking'. We used both techniques for all the baselines in both languages, as explained in Section 4.1 and Section 4.2.
>
> We have proved that compared with the best of both worlds, i.e., single grained models ('Our BERT models'), the multi-grained model (AMBERT) works better.

---

### Official Review · AnonReviewer2 · 2020-10-28
**Novel and effective multi-grained BERT**

**Rating:** 7
**Confidence:** 3

**Review:**

Summary:

This paper proposes a novel Transformer-based architecture AMBERT which uses the multi-grained characteristics of several natural languages. The novelty of the work relies on two encoders sharing parameters: one focusing on a fine-grained representation of the text (characters or words) and the other one on a coarse-grained representation of the text (words or n-grams). Both fine-grained and coarse-grained representations are used during pre-training and fine-tuning. The architecture, which combines fine-grained tokenized representations with coarse-grained tokenized representation, achieves very strong performance on benchmark datasets for Chinese and English, outperforming BERT and other Transformer-based models most of the time. The authors also provide ablation studies and analyses using variations of AMBERT: AMBERT-Combo (fine-grained and coarse-grained encoders but without parameter sharing) and AMBERT-Hybrid (one encoder using the concatenation of fine-grained and coarse-grained representations).


Pros:
-	The architecture is novel and take advantage of the multi-grained components within the sentences (character, word, n-gram) while sharing parameters between the two encoders and thus improving efficiency.
-	The architecture’s performance is validated empirically on many tasks and datasets. AMBERT outperforms most of the time recent Transformer models on both Chinese and English benchmark datasets.
-	Ablation studies in the form of a comparison with similar architectures (Combo or Hybrid), analyses of attention maps and distance between representations are proposed and help understand the performance and differences between models


Cons:
-	Even though the performances are almost always significantly better than BERT or other Transformer models, the cost is also a significant increase in the number of parameters to train (108M for Google BERT vs 176M for AMBERT in Table 1 and 110M vs 194M parameters in Table 4)


Questions:
-	Could the authors provide more information regarding the coarse-grained tokenization algorithm used?
-	Regarding the last paragraph of section 4.5, could the estimated quantities be biased/inaccurate if the 10K sampling is performed only once?



Minor Comments:
-	There is a typo at the end of the second paragraph of section 1: "tokeniztion" instead of "tokenization"
-	It would be interesting to see the study generalized to other scales of the input documents (sub-words, pairs of characters, pairs of phrases etc.)

---

> ### Author Response · Authors · 2020-11-23
> **Response to AnonReviewer2**
>
> Thank you for your review.
>
> For your cons:
> 1. Even though the performances are almost always significantly better than BERT or other Transformer models, the cost is also a significant increase in the number of parameters to train (108M for Google BERT vs 176M for AMBERT in Table 1 and 110M vs 194M parameters in Table 4)
>
> Note that the number of parameters does not increase so much, compared with BERT (less than a 15% increase compared with the coarse-grained model), because the parameters in the two encoders in AMBERT are shared. There are only additional parameters from multi-grained embeddings, which are orthogonal to those powerful parameters in Large models such as layers, heads, and hidden states. In the following parts, we will prove that with the same inferring time AMBERT still outperforms single-grained models.
>
> For the questions:
> 1. Could the authors provide more information regarding the coarse-grained tokenization algorithm used?
>
> In English, as explained in Section 4.1, coarse-grained tokenization is conducted as follows. First, a statistic language model is trained with Wikipedia data. Then, we calculated frequencies and conditional probabilities of bigrams of words and trigrams of words. With thresholds of frequencies and conditional probabilities, we select a coarse-grained vocabulary of phrases with a reasonable size. Finally, using the vocabulary, we employ a left-to-right search algorithm to perform phrase-level tokenization on the texts.
>
> In Chinese, the segmentation tool is in-house implemented with standard techniques such as those in Jieba.
>
> 2. Regarding the last paragraph of section 4.5, could the estimated quantities be biased/inaccurate if the 10K sampling is performed only once?
>
> No, we do not think so. In section 4.5, we chose Chinese sentences from CMNLI and English sentences from MNLI. We repeat this experiment on the other eight datasets, and the results are shown in the following table. The conclusions, 'about half of the tokens are fine-grained and half are coarse-grained in Chinese' and 'there are much less coarse-grained tokens than fine-grained tokens in English', are consistent. We will add the table in the revised version.
>
> |  Datasets   | Chinese words  |  Chinese total   | word rate  |English phrase  |  English total   | phrase rate  |
> |  :----:  | :----:  | :----:  | :----:  | :----:  | :----:  | :----:  |
> | CMNLI/MNLI (in paper)  | 157,511 |335,187 |47.0% |43,661 |318,985 |13.7% |
> | TNEWS/QNLI | 71,636 |137,965 |51.9% |59,506 |395,681 |15.0% |
> | AFQMC/QQP  | 94,439 |165,847 |56.9% |35,256 |237,731 |14.8% |
> | CSL/SST-2 | 836,9761 |1,739,954 |48.1% |9,757 |103,048 |9.47% |
> | CHID/CoLA  | 958,893 | 1,763,507 | 53.4% | 10,491 | 82,353 |12.7% |
> | Avg.  | - | -| 51.5% | - | - |13.1% |

---

### Official Review · AnonReviewer4 · 2020-10-29
**Interesting model proposal but not experimentally rigorous enough**

**Rating:** 4
**Confidence:** 4

**Review:**


This paper introduces AMBERT, a general purpose pre-trained model that uses both fine-grained and coarse-grained tokenizations of the sentence. Given a sentence, AMBERT tokenizes it with both vocabs, then each sequence is passed independently through a shared transformer block. During pre-training, BERT’s masking function is applied at the coarse-level, and the corresponding tokens are also masked at the fine-grained level. During fine-tuning, each tokenization leads to a sentence representation that is used for a classification loss. This classification loss comprises an agreement term, incentivizing the classifications based on each representation to be close. The goal of this paper is to explore whether using multiple levels of granularity for tokenization can lead to better models.
The authors investigate the effectiveness of this method in the monolingual setting, proposing an English and a Chinese version of the model.

### Pros:
* The question of how to best tokenize sentences and whether multiple tokenizations of the same sentence are useful has been understudied in the large language models realm. This paper draws attention to this question and starts to shed some light onto it.
* The authors have evaluated on a wide range of tasks and on both Chinese and English tasks.
* The method seems to perform well overall and better than the two proposed baselines (Combo/Hybrid). However, some
comparisons to other systems are more unfair. (see cons) Results on CLUE are more compelling than their GLUE counterpart.

### Cons:

* It is not clear on what metric the authors purport this method to improve on other ways to improve accuracy in models. The authors highlight in the abstract that the model “outperforms the existing best performing models in almost all cases”, making it a key contribution of their paper. However:
    * The proposed method adds significant overhead at all steps (pretraining/fine-tuning/inference) since the transformer block is run twice on every sentence. A fairer “Our BERT” baseline would be trained for twice longer or use an ensemble of two BERTs. All the more concerning is that this baseline is not far off from the performance of AMBERT on many English tasks.
    * The authors highlight that their method does well among methods of less than 200m parameters, but this does not seem to be a very good basis for comparison:

      * Most methods do not share parameters between two encoders like AMBERT, requiring significantly less compute. ALBERT has shown in the past that the # of parameters in a model could be significantly reduced at the cost of more compute.
      * The authors do include ALBERT, but they do so inconsistently. For Chinese, ALBERT x-large is included while for English ALBERT base is included. This is a surprising choice, especially considering that ALBERT x-large/large do better than AMBERT. It would be better to include a consistent ALBERT model and highlight that ALBERT uses more computer/param due to layer sharing.
      * The threshold of 200m parameters is arbitrary and AMBERT is significantly closer to it than most baselines.

  * The numbers reported in the external baselines are often not ideal/misleading. Take the case of the ELECTRA model in table 6 for instance. The authors choose to use the ELECTRA-base number in the paper instead of the ELECTRA-base++ one just below (see Table 8 in the Electra paper). ELECTRA-base uses 1/16th of the steps \* batch_size \* seq_length that the AMBERT model shown uses.  ELECTRA-base++ still uses 1/4 the steps \* batch_size \* seq_length that AMBERT uses. ELECTRA++ performs 2.2 points better than ELECTRA on GLUE without WNLI, which puts it firmly above AMBERT on GLUE. On SQUAD (average of SQUAD 1.1 and 2.0 dev numbers), the authors report a performance of 74.8 for their own fine-tuning of ELECTRA compared to 84.775 on the paper. This makes it hard to put a lot of faith in the comparison with external systems, both for numbers copied from the papers and from numbers obtained through fine-tuning.
  * For CLUE, the results are more impressive. It is not entirely clear to me whether the data augmentation mentioned for some tasks is used for all the models (including external ones) or only for AMBERT and its baselines. I am less familiar with this benchmark so cannot comment as precisely.

* “Note that the number of parameters in AMBERT is comparable to that in BERT because of the parameter sharing.” -> This ignores the added ~50m+ parameters added due to the additional word embedding table, which represent ~50% of BERT Base’s total parameters.
* Ablations: Though computational requirements are an obvious constraint that limit the number of experiments that can be done, key ablations that would better highlight the effectiveness of the proposed method are missing:
  * The authors mention that many approaches have used coarse-grained approaches to masking and obtained improvements, particularly BERT-wwm and SpanBERT. They implicitly use a similar masking for AMBERT (since they use the masks derived from the coarse-grained for both the coarse-grained and fine-grained representations) but their BERT baseline uses the subpar wordpiece masking.
  * Using the same masked tokens for both tokenizations of the sentence might be subpar for the AMBERT-Hybrid model, limiting how it can learn interactions between coarse and fine-grained representations.
  * It would be good to ablate the impact of the agreement regularization term. This should be easier than other ablations as it is fine-tuning specific. The use of this additional hyperparameter with several values (1 or 0, see Table 8 and 9) also means the AMBERT/AMBERT-Combo models likely had more fine-tuning runs than their BERT baseline counterparts.
* Given the high variances of fine-tuning on small data GLUE tasks such as RTE/COLA/MRPC (see for instance STILTs Phang et al, Revisiting Few-sample BERT Fine-tuning Zhang et al), it would have been good to have more than a single run for those.
* There is little discussion of previous work that uses different levels of tokenization, such as models using both character and word-based inputs. There is also no discussion of works that allocate additional tokens/representations for entity-like multi token expressions (such as those obtained in a coarse-grained representation such as “New York”). Some examples of the later include KnowBERT, Entities as Experts (the authors do mention ERNIE, but mostly for its masking function, not the use of additional entity representations). This is especially relevant since many of the phrases shown in Appendix E are indeed entity-like.
* The approach is quite brute force. In reality, the differences between the two tokenizations can be small (see for instance the examples shown in Appendix E), so re-running the entire model seems inefficient. It should be possible for instance to tokenize Sentence 1 Example 1 in Appendix E in an approach like Ernie or E-BERT (Poerner et al):
    * What Star Trek episode has a nod to Doctor Who? Star_Trek Doctor_Who (Ernie style)
    * What Star Trek / Start_Trek episode has a nod to Doctor Who / Doctor_Who ? (Ebert style)

Other concerns:
* I was not able to find anywhere whether English version of the model uses a cased or uncased vocabulary. This is important to mention in a paper that proposes to study tokenization.
* It was not entirely clear to me whether the 70k vocab is a superset of the 30k one. If not, how can you ensure that the masked tokens are aligned for both sentences?

Minor:
Section 4.2: Mix-precision -> mixed precision

### Recommendation:

In light of the above comments, **I believe this paper should be rejected**. There are important flaws in the presentation of the results and comparisons, both to external models and baselines. When correcting for those, the results are not as impressive and the claims of “SOTA” that are central to the paper are more dubious.

I believe the paper could be made stronger by trying to assess what are the changes introduced by using multiple levels of tokenization. There are interesting observations of the differences in impact it has across tasks (seems to be more useful for RC), and across languages (seems to be much more useful for English), etc. There are also likely less brute-force ways of introducing multiple levels of tokenization that would offer better complexity trade-offs, as suggested in the last con.

---

> ### Author Response · Authors · 2020-11-23
> **Response to AnonReviewer4 PART 2**
>
>
> 3.3 It would be good to ablate the impact of the agreement regularization term.
>
> Now we added an experiment. The following table shows the result of ablations for the agreement regularization term on dev sets of CLUE and GLUE (the default metric is acc.). For most tasks, the regularization term is necessary. We did not use the best value for each task and want to give uniform guidance. Therefore, for all the tasks except RACE, we suggest using an agreement regularization term of 1.0. The best practice of RACE is setting the agreement regularization term to 0.0.
>
> |  CN-Models   | dis_reg  | TNEWS  |  IFLYTEK   | CLUEWSC2020  | AFQMC   | CSL  | CMNLI   | ChID  | $C^3$  | - |
> |  :----:  | :----:  | :----:  | :----:  | :----:  | :----:  | :----:  | :----:  | :----:  | :----:  | :----:  |
> |  AMBERT   | 1.0  | **68.1** | 60.1  |  **81.6**   | 74,7  | **85.6**   | **82.3** | **87.1**  |  **69.2**  | - |
> |  AMBERT   | 0.0  | 67.9 | **60.3**  |  80.9   | **75.4**   | 85.0  | 81.1  |  86.5  | **69.2** | - |
>
> |  EN-Models   | dis_reg  | CoLA (Matthew's Corr.) |  SST-2   | MRPC  | STS-B   | QQP  | MNLI  |  QNLI  | RTE  | RACE |
> |  :----:  | :----:  | :----:  | :----:  | :----:  | :----:  | :----:  | :----:  | :----:  | :----:  | :----:  |
> |  AMBERT   | 1.0  | **61.7** | **94.3**  | **92.3**  | **55.0**   | **91.2**  | **86.2**  |  **91.3**  | **70.2**  | 66.6 |
> |  AMBERT   | 0.0  | 61.5 | 93.4  |  90.1   | 54.5  |91.1  | 85.5  |  91.2  | **70.2**  | **66.8** |
>
> 4. Given the high variances of fine-tuning on small data GLUE tasks such as RTE/COLA/MRPC
>
> These datasets are very small and sensitive to hyper-parameters. Therefore, we fixed the hyper-parameters and chose the first run of all the baselines to fairly compare the results. Now we conducted additional experiments of five random runs on the three tasks. The results on the dev sets are shown in the following table, and AMBERT-0 corresponds to the results reported in the manuscript. We can see that 1) the results in the paper are not the best ones on development sets. 2) the first run does not fluctuate much on expectation and average score.
>
> |  Models   | CoLA (Matthew's Corr.)  | RTE (Acc.)   | MRPC (F1)  | MRPC (Acc.)   | Avg. |
> |  :----:  | :----:  |  :----:  | :----:  |  :----:  | :----:  |
> | AMBERT-0  | 61.7 |  70.2 |  89.2 |  92.3 |  78.4 |
> | AMBERT-1  | 61.6 |  69.1 |  **89.5** |  **92.5** |  78.2 |
> | AMBERT-2  | 61.0 |  69.1 |  88.2 |  91.6 |  77.5 |
> | AMBERT-3  | 61.4 |  **71.0** |  88.7 |  91.9 |  78.3 |
> | AMBERT-4  | **62.9** |  70.2 |  88.7 |  92.0 |  **78.5** |
> | Expectation  | 61.7 |  69.9 |  88.9 |  92.1 |  78.2 |
>
> 5. There is little discussion of previous work that uses different levels of tokenization, such as models using both character and word-based inputs.
>
> First, the differences between AMBERT and the existing models are significant. The key question is not whether or not to use coarse-grained tokens, but is how to do it. Note that ERNIE stated that in their paper, "ERNIE is designed to learn language representation enhanced by knowledge masking strategies". Compared with ERNIE, AMBERT leaned additional language representations for coarse-grained tokens including not only entities but also other phrases such as "based on" in English and "打死(beat to death)" in Chinese which can be found in our case study.
>
> Second, the other works that use tokens/representations for entity-like multi-token expressions such as E-BERT or KnowBERT, were only proved effective in unsupervised question answering, relation extraction, entity linking, entity typing, and word sense disambiguation. All the downstream tasks are sensitive to entities. However, these knowledge/entity enhanced methods are not proved to be so effective in generalized NLP tasks.
>
> 6. The approach is quite brute force. In reality, the differences between the two tokenizations can be small
>
> We will explore other methods for multi-grained pretraining. We leave the work as a future study. What we want to claim in this paper is that multi-grained tokenizations are useful to train a powerful pre-trained model.
>
> 7. others
>
> 7.1 I was not able to find anywhere whether English version of the model uses a cased or uncased vocabulary. This is important to mention in a paper that proposes to study tokenization.
>
> We used uncased vocabulary for all the baselines. We will mention it.
>
> 7.2 It was not entirely clear to me whether the 70k vocab is a superset of the 30k one. If not, how can you ensure that the masked tokens are aligned for both sentences?
>
> The 70k vocab is a superset of the 30k set.
>
> 8. There are interesting observations of the differences in the impact it has across tasks (seems to be more useful for RC), and across languages (seems to be much more useful for English), etc.
>
> Note that AMBERT is more useful in Chinese, which has been discussed in Section 4.5.

---

> ### Author Response · Authors · 2020-11-23
> **Response to AnonReviewer4 PART 1**
>
> Thank you for your review.
>
> 1. It is not clear on what metric the authors purport this method...
>
> 1.1 The proposed method adds significant overhead at all steps...
>
> We have demonstrated in our experiments that the multi-grained model, AMBERT, is better than single-grained models, with comparable parameters (10-50% increase).
>
> We have conducted additional experiments in which we use a simplified AMBERT, AMBERT-Single, with only one single-grained encoder used in inference. The results show that AMBERT-Single can perform almost equally well as the full AMBERT, but with comparable inference time as the single-grained baselines. We will include the results in the revised manuscript. Please also refer to our responses to reviewer5.
>
> Further note that AMBERT and Our BERT baselines are fed into the same training examples, and we found in experiments that the training of AMBERT is not two times longer. As for twice training of BERT, which you point, it is close to AMBERT-Combo, which we compared.
>
> 1.2 The authors highlight that their method does well among methods of less than 200m parameters...
>
> We do not compare our model with Large models because we have limited computation resources to pre-train Large models such as AMBERT large. We think that our findings on multi-grained modeling, etc are orthogonal to the increasing of layers in modeling, and thus they have values for future research in the area.
>
> In Chinese, we include all the base models including XLNET-mid in the CLUE leader-board with 200M parameters. Basically, 200M is a threshold to filter out large models. In English, we compare our models with all the base models.
>
> 1.3 The numbers reported in the external baselines are often not ideal/misleading.
>
> First, in ELECTRA paper Appendix B, they indicated that：
> "For the GLUE test set results, we apply the standard tricks used by many of the GLUE leaderboard submissions including RoBERTa (Liu et al., 2019), XLNet (Yang et al., 2019), and ALBERT (Lan et al., 2019). Specifically: a) For RTE and STS we use intermediate task training; b) For each task we ensemble the best 10 of 30 models fine-tuned with different random seeds."
> We did not use any tricks in our finetuning. In fact, we used the finetuning method, HuggingFace Transformer, for all the baselines.
>
> Second, for the model ELECTRA, we used the google released model, "google/electra-base-discriminator". The performance of our running is similar to the 'ELECTRA-base' results reported in the paper. Besides, the results of SQuAD reported in their paper are achieved with an additional question-answering module. Therefore, we run ELECTRA by ourselves with HuggingFace Transformer.
>
> Third, the key point in this paper is that using multi-grained tokenizations is better than single grained tokenization for pre-trained models. We have conducted fair experiments to compare the two approaches. Besides, we found that even under the basic settings of BERT-base, AMBERT can also achieve comparable or better performance than the baseline models.
>
> 1.4 For CLUE, the results are more impressive.
>
> Data augmentation mentioned for the tasks is used for all the models. The comparisons are fair.
>
> 2. This ignores the added ~50m+ parameters.
>
> Yes, it is more accurate to say that the added parameters are the additional word embeddings with a 10-50% increase, as you point out. We will revise the explanation.
>
> 3. Ablations
>
> 3.1 their BERT baseline uses the subpar wordpiece masking
>
> Our BERT baselines used the same masked data with AMBERT, which means there is no difference in the wordpiece masking method. Also, to make fair comparisons, we kept all the settings the same between AMBERT and the BERT baselines.
>
> 3.2 might be subpar for the AMBERT-Hybrid model
>
> If we do not mask the same tokens for both tokenizations of AMBERT-Hybrid, there will be an information leak between fine-grained tokens and coarse-grained tokens, which will make MLM too easy.

---

### Official Review · AnonReviewer5 · 2020-11-06
**Require better comparison.**

**Rating:** 5
**Confidence:** 5

**Review:**

The paper combines fine and coarse-grained tokenizations to learn word and phrase-level representations. The authors introduce two variations on AMBERT: (1) using two separate encoders for fine and coarse, and (2) combine fine and coarse into a single encoder. This method is more expensive in terms of computations. The improvement is incremental, but it seems very effective in representing Chinese. However, the motivation for adding fine and coarse granularities is not well introduced.

Strengths:
- The performance is consistently good, although AMBERT has more parameters than baselines: BERT and ALBERT.

Weaknesses:
- The model is more complex and computationally more expensive (2-4x)
- The proposed method seems not very effective for English, only to Chinese
- Lack of intuition

Questions and Suggestions:
- Can you train a model with the same parameters as the baselines? I think it would show the significance of the approach.
- Please elaborate more about the intuition of the proposed method

** Post-Rebuttal **

> I want to thank the author for addressing my concerns. I will keep my score. The paper has merits, but the comparison is not fair since they have different parameters with the baselines unless they have smaller parameters like ALBERT.

---

> ### Author Response · Authors · 2020-11-23
> **The intuition and the additional experiments about computation.**
>
> Thank you for your review.
>
> 1. Can you train a model with the same parameters as the baselines?
>
> Yes, we have done it, which supports our claims. We conducted additional experiments on CLUE/GLUE/SQuADs/RACE with the same amount of inference computation for AMBERT and the Our BERT baselines. In the AMBERT variant, we keep the same processes for pretraining and finetuning and conduct inference with only one single-grained encoder. The results on dev sets are shown in the following table, from which we can conclude that a) for the English tasks, AMBERT-Single achieves similar results as AMBERT and outperforms "Our BERT (Single)" with a large margin using the same inference time. b) for Chinese tasks, AMBERT-Single is slightly worse than AMBERT, and performs much better than "Our BERT (Single)". Therefore, in practice, one can train an AMBERT with two encoders and use only one of them in inference. This simple method will give better accuracies with the same amount of inference computation.
>
> The results in the following table are development set performance of the CLUE, GLUE, SQuAD and RACE with AMBERT-Single or Our BERT (better one) for inferring. CN-Models means Chinese pre-trained models and EN-Models represent English ones. CoLA uses Matthew's Corr. We report EM of CMRC2018 and the average EM of SQuAD1.1 and SQuAD2.0. The other metrics are all accuracies.
>
> |  CN-Models   | Speedup  | Avg. | TNEWS  |  IFLYTEK   | CLUEWSC2020  | AFQMC   | CSL  | CMNLI  |  CMRC2018  | ChID  | $C^3$ | - |
> |  :----:  | :----:  | :----:  | :----:  | :----:  | :----:  | :----:  | :----:  | :----:  | :----:  | :----:  | :----:  |:----:  |
> |  AMBERT   | 1.0  | 75.3 | 68.1  |  60.1   | 81.6  | 74.7   | 85.6  | 82.3  |  68.8  | 87.1  | 69.2 | - |
> |  AMBERT-Single   | 2.0x  | 74.8 | 68.0  |  59.5   | 81.3  | 74.2   | 85.5  | 82.1  |  67.4  | 86.6  | 68.5 | - |
> |  Our BERT (Single)   | 2.0x  | 73.4 | 67.8  |  58.7   | 79.0  | 74.1   | 84.5  | 80.8  |  65.5  | 83.4  | 66.7 | - |
>
> |  EN-Models   | Speedup  | Avg. | CoLA  |  SST-2   | MRPC  | STS-B   | QQP  | MNLI  |  QNLI  | RTE  | SQuAD | RACE |
> |  :----:  | :----:  | :----:  | :----:  | :----:  | :----:  | :----:  | :----:  | :----:  | :----:  | :----:  | :----:  |:----:  |
> |  AMBERT   | 1.0  | 79.0 | 61.7  | 94.3 | 92.3  | 55.0   | 91.2  | 86.2  |  91.3  | 70.2  | 80.9 | 66.8 |
> |  AMBERT-Single   | 2.0x  | 78.9 | 62.2  |  93.2   | 92.5  | 55.0   | 91.2  | 86.1  |  91.4  | 70.6  | 80.3 | 66.7 |
> |  Our BERT (Single)   | 2.0x  | 77.1 | 56.6  |  92.4   | 89.7  | 54.2   | 90.4  | 85.1  |  90.6  | 69.1  | 80.2 | 62.6 |
>
> 2. Please elaborate more about the intuition of the proposed method.
>
> First, it seems better to keep both coarse-grained tokens and fine-grained tokens for language processing particularly for Chinese. The fact that the use of word boundary information is important for Chinese has been demonstrated in other work [1-5]. AMBERT adopts a simple way to utilize word boundary information with both fine-grained tokens and coarse-grained tokens as input.
>
> Second, the parameters in the two encoders of AMBERT are shared. It means that the ways of combining representations of fine-grained tokens and the ways of combining representations of coarse-grained tokens *in the same contexts* are exactly the same. In other words, it uses the same ways to combine similar tokens no matter whether they are fine-grained or coarse-grained.
>
> Third, the results of AMBERT are better than those of AMBERT-Hybrid and AMBERT-Combo. It indicates that simply combining a fine-grained model and a coarse-grained model does not work. The objective of pre-training in AMBERT also has an effect of making representations of similar fine-grained tokens and representations of similar coarse-grained tokens get closer.
>
> We will add this intuitive explanation into the revised version.
>
> ref:
>
> [1] Xiao, Dongling, et al. "ERNIE-Gram: Pre-Training with Explicitly N-Gram Masked Language Modeling for Natural Language Understanding." arXiv preprint arXiv:2010.12148 (2020).
>
> [2] Levine, Yoav, et al. "PMI-Masking: Principled masking of correlated spans." arXiv preprint arXiv:2010.01825 (2020).
>
> [3] Li, Xiaonan, et al. "FLAT: Chinese NER Using Flat-Lattice Transformer." arXiv preprint arXiv:2004.11795 (2020).
>
> [4] Diao, Shizhe, et al. "ZEN: pre-training chinese text encoder enhanced by n-gram representations." arXiv preprint arXiv:1911.00720 (2019).
>
> [5] Zhang, Yue, and Jie Yang. "Chinese ner using lattice lstm." arXiv preprint arXiv:1805.02023 (2018).

---

### Author Response · Authors · 2020-11-23
**summary of all the responses and the changes in our revised paper**

Thank you for all your reviews and valuable comments. We first give a summary of the responses and changes in our revised manuscript.

1. Parameters and computation

Note that the number of parameters does not increase so much, compared with BERT (less than a 15% increase compared with the coarse-grained model), because the parameters in the two encoders in AMBERT are shared. There are only additional parameters from multi-grained embeddings.

As for the computations, we conducted additional experiments in which we use only the fine-grained encoder in AMBERT in inference, denoted as AMBERT-Single. With similar inference time, AMBERT-Single still performs much better than single-grained models in both English and Chinese. Please refer to detailed results in our response to reviewer5.

2. Validity of experiments

We have done rigid experiments to compare AMBERT and single-grained pre-trained models fairly. We keep all the settings the same to AMBERT and the baselines including masked tokens and all of the hyper-parameters. The only difference between AMBERT and our BERT baselines is the architecture. We, thus, can conclude that the multi-grained pre-trained model is better than single-grained ones.

3. Intuition explanation of AMBERT

First, it seems better to keep both coarse-grained tokens and fine-grained tokens for language processing particularly for Chinese.

Second, the parameters in the two encoders of AMBERT are shared. It means that the ways of combining representations of fine-grained tokens and the ways of combining representations of coarse-grained tokens *in the same contexts* are exactly the same.

Third, the results of AMBERT are better than those of AMBERT-Hybrid and AMBERT-Combo. It indicates that simply combining a fine-grained model and a coarse-grained model does not work.

4. Novelty of work.

We find that with the architecture one can achieve better performances than the existing models. This cannot be easily found and was not known so far. There are also comprehensive experiments on the comparisons. All of these should be valuable for future research. We think, therefore, that our work represents a significant contribution.

Our revised version has the following changes.

1. We add intuitive explanations.
2. The additional experiments for inferring time are included.
3. We add more ablations studies about regularization term and coarse-grained token rate.

---

### Decision · Program_Chairs · 2021-01-07
**Final Decision**

**Decision:**

Reject

**Comment:**

This paper suggests extending pre-trained contextual language models to use both fine-grained and coarse-grained tokenizations of the sentence. A sentence is tokenized twice and then each is passed through a transformer block, with shared parameters except the embeddings. Having 2 granularities shows gains.

Pros

- Easy to read paper, straightforward method
- Gets experimental gains from using word/phrase combo
- Evaluates on a range of tasks

Cons

- Novelty is limited, since other models like SpanBERT and ZEN already explore different tokenization granularities
- Improvements may come as much from the ensemble of two models as the two tokenization granularities
- Number of parameters or amount of computation are increased by method, though authors do significantly address this in their revised paper.
- Some over claiming of results when there are modest incremental gains on small models (the abstract sentence "outperforms the existing best performing models in almost all cases" suggests that we are going to get results of a new model outperforming the state of the art models on tasks, but really we get improvements over baseline models at the BERT-base size. I believe this is fine for experiments to show the scientific value of ideas but it should not be described as it presently is in this abstract.
- Gains are more for Chinese than English

On the better results for Chinese: Isn't the reason that the results are more impressive for Chinese because in Chinese the fine-grained version is just single characters, but this is more fine-grained than standard BERT word pieces in English, where the word pieces are already commonly words, most of which would be two or more character sequence in Chinse (whether for common words like, say "fishing" or "vault" or place names like "Mississippi"), so the fine-grained Chinese here is more fine-grained than the standard English wordpieces, and so not too surprisingly there are bigger gains from using the Chinese word segmenter granularity. But really this is sort of equivalent to how the original BERT authors showed that you could get gains by masking whole words not individual word pieces. And at any rate, the value of word segmentation for Chinese was already shown by Yiming Cui et al.'s paper on Chinese BERT, no?

Overall the strong majority of reviewers were unconvinced that this paper was suitable for ICLR 2021. They mainly emphasized concerns of novelty, missed or unfair comparisons, concerns of extra parameters or computation, and the fact the paper is somewhat incremental. I would add to that that to the extent that this paper is primarily an examination of the value of different granularities, that feels much more like a linguistic question for an NLP conference than an ML question well suited in particular to a conference on learning representations like ICLR. That is, the choice of granularities is hand-specified, and/or the grouping is done by simple n-gram statistics, not by learning representations. As such, I do not think the paper should be accepted to ICLR at this time, and in general think that an NLP venue may be more appropriate for it.